# The Most Common Handball Injuries: A Systematic Review

**DOI:** 10.3390/ijerph191710688

**Published:** 2022-08-27

**Authors:** Helena Vila, Andrea Barreiro, Carlos Ayán, Antonio Antúnez, Carmen Ferragut

**Affiliations:** 1Department of Special Didactics, Faculty of Education and Sport Sciences, University of Vigo, 36005 Pontevedra, Spain; 2Faculty of Education and Sport Science, University of Vigo, 36310 Vigo, Spain; 3Research Group in Optimization of Training and Sports Performance (GOERD), University of Extremadura, 10005 Caceres, Spain; 4Department of Biomedical Sciences, Faculty of Medicine and Science Health, University of Alcala, 28801 Alcalá de Henares, Spain

**Keywords:** athletes, team sports, prevention, epidemiology, overuse injuries

## Abstract

Handball is a team sport involving a great physical demand from its practitioners in which a high number of injuries occur, affecting individual and collective performance. Knowledge of the injuries is of great importance for their prevention. The objective of the present study was to identify, locate and compare the most frequent injuries and injury mechanisms in handball practice. It was carried out following the Preferred Informed Item for Systematic Reviews and Meta-analysis (PRISMA) guidelines. The source of data collection was direct consultation of the PubMed and Medline databases. Several keywords were used for the documentary retrieval, and the quality of the studies that were selected was evaluated. Of the 707 studies retrieved, only 27 were considered appropriate for the review, and quality scores were obtained that ranged from 10 to 26 points, out of a maximum of 28. The most frequent injuries in handball players are located in the lower limbs (thigh, knee and ankle), and in the shoulder in the upper limbs. Regarding the playing position, the players who play over the 6-m line are the most affected by injuries, while the women players have a higher probability of injury. Most injuries occur during competition.

## 1. Introduction

Handball is a team sport involving a high physical demand from its players, as, during the game, there are accelerations, changes of direction, throws, jumps and numerous contacts between players [1,2], which cause this sport to present a high incidence of injury [3].

The injury rate in young players has been established between 9.9 and 41.0 injuries per 1000 match hours and between 0.9 and 2.6 per 1000 training hours [3]. In the case of elite senior players, this number is increased by the occurrence of recurrences of old injuries and overuse injuries [3,4]. As a result, the International Olympic Committee (IOC) surveillance system ranks it as one of the Olympic sports with the highest injury rates [5]. This is why injury prevention in young handball players is one of the priorities for researchers [1].

It is important to be able to have relevant information to be up-to-date on the different aspects related to the game and training [6], and systematic reviews allow access to the latest studies and advances made in the scientific field. In a systematic review focused on doctoral theses defended in handball [6], it can be observed that the section on injuries is not a topic that has attracted much attention in this sport. However, in recent years, systematic reviews on handball have been identified, focusing on aspects of the game and performance [7,8] and on injury profiles [9,10].

In this line of research, the first aspect that must be addressed with the aim of preventing injuries, is the identification of the injury profile of the players, for which it is essential to know the most frequent injuries associated with the practice of this sport. This will allow us to identify the risk factors associated with an injury, to know the possible mechanisms that produce it, as well as to determine whether the specific playing position, age or sex are associated in a particular way with one type of injury or another. In this way, it will be easier for coaches to design training tasks focused on the prevention of these injuries, improving the performance of the player and the team, as well as extending the sporting life of these professionals.

For all these reasons, the aim of this study was to identify, locate and compare the most frequent injuries and injury mechanisms in handball.

## 2. Materials and Methods

### 2.1. Search Strategy

This systematic review was conducted following the Preferred Reported Item for Systematic Reviews and Meta-analyses (PRISMA) standards [11]. The source of data collection was direct consultation and access, via the Internet, to the scientific literature library of the PubMed and Medline databases. For document retrieval, several keywords were used. Five searches were carried out in different databases. The Boolean operator “AND” was used to combine the descriptors. The descriptors “prevention injuries” AND “handball” were used for the first search equation. For the second search equation, the word “injuries” was changed to “injury”, with the words used being “prevention injury” AND “handball”. Finally, in the third search equation, the term “prevention” was eliminated. In this last search the words used were “handball injuries”. The last two search equations used the search terms “prevention injuries AND handball” and “handball injuries”. A search was performed for each possible combination of the above keywords in each database used.

### 2.2. Selection Criteria

A temporal selection of 10 years was used and only articles written in English and registered as original articles were included. Only articles that included larger samples of nine male and nine female handball players, and that included a definition of injury (Table 1), were selected. All articles were excluded if they did not discuss handball, were not written in English, were reviews and if the term injury was not defined.

### 2.3. Data Extraction and Methodological Design

Information on the characteristics of the injuries, the intervention program, and the results were extracted by reading the original studies, assessing the type of design, the sample, the aims and the protocols carried out.

Moreover, the quality of the selected studies was assessed by means of questions [12]. Each answer was given a score, with “Yes” being scored with 2, “N/P” with “1” and zero for the answer “No”. The answer “N/P” indicates that the study does refer to what is asked but appears in a generalized way for several sports or does not appear in a specific way for handball and it is not possible to ascertain this information. Therefore, a maximum score of 28 points was established, with the articles closest to this score being those with the highest score quality. Data extraction and methodological evaluation were carried out qualitatively by three evaluators. In cases where a dispute arose, it was decided among the three evaluators by consensus agreement [13].

## 3. Results

### Selected Studies

Of the 707 studies retrieved in the literature search, only 27 were deemed appropriate for the review (Figure 1).

The 27 studies obtained quality scores ranging from 10 to 26 out of a maximum of 28 points (Table 2). The main characteristics of each of the selected articles are listed in Table 3. The largest samples refer to professional players.

## 4. Discussion

### 4.1. Epidemiology and Relationship to Injury Definition

In the review of the studies, a great diversity of injury definitions was found, which was one of the handicaps when addressing an epidemiological profile of handball injuries. Of the 27 articles selected to be part of the review, 12 of them have a similar definition (in a broad sense), each with its own nuances. Some of the authors [3,5,16,23,26,30,33] define injury by referring to “a physical complaint”. Others such as Aman et al. [28] and Aman et al. [32] use the words “physical damage” to describe injury. Giroto et al. [31] in their article use “musculoskeletal pain”. Anderson et al. [27] and Clarsen et al. [35] make use of the adjectives “pain, distress, stiffness, instability and weakness” to give their definition. However, other articles present more open definitions. This is the case of Ruehlemann et al. [20] (2019) and Monaco et al. [22] define injury as damage that causes absence in training or competition. In the case of Asker et al. [2] and Aasheimen et al. [1] we are told that it causes a moderate or severe reduction in training volume, performance or even total capacity. In the case of Luig et al. [17] defines contact injury as any injury due to external forces and non-contact injury as injury without contact with another player or object.

Considering the different injury definitions used for data analysis, a rigorous epidemiological profile cannot be drawn up, but in general, the highest number of injuries occur in the lower extremities, but without reaching a consensus on which has the highest incidence. In relation to upper limb injuries, shoulder injuries are the most frequent. It seems that the greatest number of injuries are caused by contact with another player, without identifying whether they occur more during competition or during training.

### 4.2. Injury Location in Handball

Handball is among the sports with the highest injury rates (82.2%) within Olympic sports [18]. Most authors identify the greatest number of injuries in the lower extremities (LES), highlighting the knee, thigh and ankle areas, but without reaching a consensus on which has the highest incidence. On the other hand, shoulder injury is the most common upper extremity (UE) injury.

In a study conducted at the World Men’s Handball Championship (MWBC) in Qatar 2015 [33], they found that 58.3% of the injuries were located in the lower extremities, mainly in the ankle, thigh and knee, while 16.7% affected the upper extremities, mainly the shoulder and fingers/thumb. In another study conducted in the same year, the knee and shoulder were shown to have a high injury rate with 20% and 22% respectively [34]. All these results were ratified with subsequent studies [14,17,21,28,30].

Along the same lines, in the 2017 French MWBC [23], they conclude that the most affected parts during this tournament were the ankle (19.3%), head/face (17.3%), knee (15.1%) and thigh (12.9%). Similar percentages to those presented by Von Rosen et al. [26]. In the study presented by Asai et al. [19], in players aged 13–14 years, the most affected body areas were the head/face, ankle, knee and wrist/hand.

### 4.3. Upper Extremities (UE) 

Starting with the shoulder, Asker et al. [15] analyzed range of motion (ROM), as well as strength, external rotation, internal rotation and scapular dyskinesia, in relation to shoulder injuries. In the analysis of ROM, they found no association between ROM and injury. Nor was a connection observed between scapular dyskinesia during flexion and the incidence of shoulder injury. However, they did find a link between scapular dyskinesia during shoulder abduction in players. This may be explained by the fact that abduction resembles the throwing action. In contrast, no such link was found in women players. What they did find was a link between a weakness between isometric external and internal rotation of the shoulder in relation to injuries in women players.

Andersson et al. [27] found no association between overuse shoulder injury and obvious scapular dyskinesia. Contrary to what was found in the study by Asker et al. [15], these authors did obtain significant results on the ROM of internal rotation, relating increased internal rotation to overuse injury. On the other hand, they observed that being a woman may represent a risk factor for this type of injury.

According to Asker et al. [2], the prevalence of any shoulder problem was higher in women and men front row players compared to those playing closer to the 6 m line. This may perhaps be explained by the fact that the second line has more interventions and greater exposure to contact. In the same vein, these authors found a higher prevalence of injury in women players than men players in the front row.

In the retrospective study on tendinopathy by Florit et al. [21], among several sports (basketball, hockey, volleyball, …,), handball was the sport with the highest incidence of tendinopathy in the upper limbs, particularly in the shoulder, probably due to a high number of throws performed at high speed.

According to Moller et al. [29], the injury rate was higher among players who increased their handball load by more than 60% compared to those players who decreased or increased their handball load by 20%.

### 4.4. Lower Extremities (LE)

One of the most severe injuries leading to a long recovery period is the anterior cruciate ligament (ACL) injury. Along these lines, Oshima et al. [25] reported that those players who sustained ACL injuries had poor static balance compared to players who were not injured, concluding that balance training is effective in preventing non-contact ACL injuries. In the study presented by Ruhlemann et al. [20], they evaluated knee stability functionality with a battery of tests, finding that stability differs depending on the position on the field. The results of this study show that the pivot is the player with the best stability values in the two-legged stability test, while in the single-legged balance, it was the wingers who obtained the lowest results. In another of the tests, in the plyometric jumping test, the pivots had the longest contact time with the ground. Finally, in the one-legged countermovement jumping test, the front row jumped more than the pivots. Therefore, the data obtained can be used for an objective assessment of knee functionality and stability.

### 4.5. Mechanism of Injury in Handball

When analyzing the mechanism of injury in handball, studies show that the majority of injuries (>50%) are caused by contact with another player [5,14,17], with traumatic injuries being the most common, followed by overuse injuries (63% vs. 37%) [3,14,21].

In the same vein, Bere et al. [33] recorded that 61.4% of injuries were caused by player-to-player contact. The remaining were non-contact (15.9%) and overuse (12.1%). However, in another study 34% of injuries were found to be caused by contact with another player, 2% in contact with the ball and 64% non-contact [30].

Regarding the most common areas of traumatic injury, the different studies analyzed agree that ankles and knees are the most injured areas, with the percentages of occurrence of ankle injuries ranging from 19.4% in the study by Giroto et al. [31] to 24% in the one presented by Moller et al. [3] and knee injuries from 13.5% in the former to 19% in the latter.

In the latter study, it was observed that the majority of injuries were new (65%), compared to 35% that were recurrent. Furthermore, in one of the age groups analyzed in this study (U16), there was a significant relationship between a previous injury and a new injury [3]. The results are in line with those presented in Japanese grassroots players [14]. However, in the study by Raya-Gonzalez et al. [10], no significant differences were observed between the First and Second Division groups in injury recurrence, mechanism and severity.

An investigation conducted on German players [17], recorded that the most affected parts in the mechanism by direct contact were the head (87.5%), the hands (83.8%), the shoulder (70.2%) and the ankle (62.9%). In injuries caused by indirect contact, the parts most affected by this mechanism were the knee (27.7%), the shoulder (27.7%) and the thigh (21.8%). Finally, in non-contact injuries the parts most affected were the thigh 56.4%, the knee 27.8% and the ankle 19%.

In the case of overuse injuries, Aasheim et al. [1] specified that this type of injury was higher in male junior players (39%), particularly in the shoulder and knee, finding a higher prevalence in each shoulder separately while the knee had a higher prevalence in the sum of both knees (dominant and non-dominant).

In another vein, Goes et al. [16] found that muscle injuries were the most prevalent, followed by joint injuries and finally tendinopathies. Within muscle injuries the back of the thigh had the highest incidence (27.8%), followed by the shoulder and the anterior thigh (both 15.8%). In joint injuries, the most common were knee (30.5%) and ankle (33.3%). Finally, in tendinopathies, the knee had the highest incidence (42.9%) followed by the shoulder (33.3%). Other studies by are along the same lines [14,21].

### 4.6. Injuries by Gender and by Category

Regarding gender, in the research presented by [3], men players in the U18 age group had a higher incidence of injury than women players in the same age group. Along the same lines, in the study presented by Asai et al. [19], significant differences in injury incidence were recorded between men (32.7%) and women players (20.1%). On the contrary, [31], observed higher recurrent injuries in women players (66.7%) than men players (33.3%). These results are in line with those presented by other authors [24,28,32].

Regarding injuries recorded during competition and training, in the study conducted on Brazilian athletes [31], a higher number of injuries were recorded in competition for women players (56.2%) compared to men players (46.3%), while during training the behavior was the inverse.

Along the same lines, women players had a higher number of injuries with more than 7 days lost compared to men players, i.e., they had more severe injuries [5]. In another study, they found that the majority of injuries, 41.4% were of moderate severity (2–7 days), but women players presented a lower percentage in this type of injury than men players [31]. Bere et al. [33], state that the injuries that caused the longest player recovery times were head/face (82.4%), knee (80%) and ankle (65.2%) injuries.

In terms of category, older players were more at risk of injury than younger players [23]. In relation to severity, the senior category had moderate severity injuries (8–28 days) with an incidence of 12.3%, followed by minimal severity injuries (1–3 days) with an incidence of 9.6% and minor injuries (4–7 days) with an incidence of 7.2%. In comparison, the U18 and U16 categories had a higher incidence of moderate severity, followed by mild and finally minimal [3]. In terms of injury severity, moderate injuries (8–28 days) had the highest incidence at 32%, followed by minor (4–7 days) at 27% and severe (>28 days) at 26% [30]. In the study presented by Mashimo et al. [14], 38.2% of the injuries had the athlete lose more than 28 days of training, and among that percentage 35.4% were produced by contact with another player.

### 4.7. Incidence of Injury by Specific Position

Regarding the incidence of injury according to playing position, it could be said that the pivot position has the highest risk of injury. As for the wingers and the second line, it is not clear which has the highest incidence.

According to Bere et al. [33] the players with the highest incidence of injury were those playing on the 6-m line, followed by the wingers, the first line and finally the goalkeepers. In the study published by Rafnsson et al. [30], it is observed that the most injured players were the goalkeepers with 67% of the injuries caused by overuse and 33% by acute injury, followed by the wingers, who were injured half by overuse and half by acute injury. Finally, the front line accounted for 38% of injuries due to overuse and 62% due to acute injury.

In terms of the position players occupy on the field, a similar incidence of injury is observed between the different playing positions, as well as categories, with the second line presenting more knee and cartilage problems [22]. However, other studies found that the players most likely to be injured are the front row followed by the 6-m players [14,23].

Luig et al. [17] go a little deeper and relate playing position to a particular injury mechanism, finding that pivots (58.4%) and wingers (56.9%) are the players with the highest proportion of injury by the mechanism of direct contact with another player (51.4% of injuries). This is followed by the front line (56.9%) and finally goalkeepers with 44.4%. Twenty-six per cent of injuries are caused by indirect contact, with the front row being the most affected (31.5%). Finally, 22.6% of injuries were non-contact, with goalkeepers having the highest percentage of injuries (48.9%).

### 4.8. Timing of Injury

Finally, there seems to be no consensus on when most handball injuries occur, whether in training or in competition. According to Engebretsen et al. [5], at the London Olympic Games, most injuries occurred during competition. The same conclusions were reached by other authors [22,26,30]. In the study of Brazilian men and women players [31], 59.9% of injuries were recorded during training compared to 48.1% during matches. In this same line of thought, the study presented by Mashimo et al. [14] also coincides, following a competitive season, in young Japanese players, a percentage of 28.4% injuries in competition were recorded, compared to 71.6% during training.

In the study presented by Raya-González et al. [10], the injury rate of the same team was analyzed during two seasons, in which they played in two different categories (first division vs. second division). A significantly higher incidence of injury was observed in the second division group in training and match play.

Analyzing the competition further, during the 2015 MWBC in Qatar [33], most injuries were recorded during the second part of the first half. However, Luig et al. [17] found no difference between the first and second half of the match. What they did find was that more injuries were identified in the last 10 min of each half, especially in the last minutes of the second half. Moreover, most of the injuries occurred in the offensive area and the area of the pitch with the highest proportion of injuries was the central area between 6 and 9 m.

### 4.9. Injury Prevention

Regarding aspects related to injury prevention, information is limited, with the warm-up being the appropriate place to work on this aspect. Andersson et al. [27] report that coaches consider that there is a risk of shoulder injury for youth handball players. These authors raise the importance of implementing an OSTRC Shoulder Injury Prevention Program as it works on several risk factors associated with shoulder injuries. The coaches consulted consider that it is a suitable program to introduce in the warm-up, but express difficulties in its use, as they consider it more relevant to use training time to work on specific content. Asker et al. [2] highlight the importance of introducing a clinical follow-up program on a routine basis and improving medical support, taking into consideration gender aspects in training stages. In aspects related to shoulder training load, Moller et al. [29], identify that rate above 60% increase the injury rate, even in youth players with normal shoulder characteristics. Youth players with scapular dyskinesia or reduced external rotation strength do not appear to be more likely to suffer a shoulder injury if the training load is not increased by more than 20%. For female players, the study presented by Oshima et al. [25] identifies the need for static balance work to reduce the risk of preventing cruciate knee ligament injuries in female players, when injuries are not caused by contact with another player. They propose that this work should be carried out during warm-up.

## 5. Conclusions

The most frequent injuries in handball players are located in the lower limbs (ankle, knee and thigh), and in the shoulder in the upper limb. In terms of playing position, the results suggest that there is a prevalence and pattern of injuries according to the specific position, with senior players playing at 6 m being the most affected by injuries. Women players are more likely to be injured. No data are presented to confirm that more injuries occur during competition than in training. However, the last 10 min or so of each of the two halves of a match seem to be the most frequent time for an injury to occur. As to whether the first or second half of the match is the time when more injuries occur, there are insufficient data to support one side or the other. Regarding injury prevention, the right time to work on it is during the warm-up phase. For cruciate ligament knee injury in female players, static balance work reduces the risk of injury.

Finally, one of the main limitations of this study is the diversity of methodologies used in the different studies, as well as the differences in the definitions of injuries and their severity. This has been one of the handicaps in approaching the review of the data.

## Figures and Tables

**Figure 1 ijerph-19-10688-f001:**
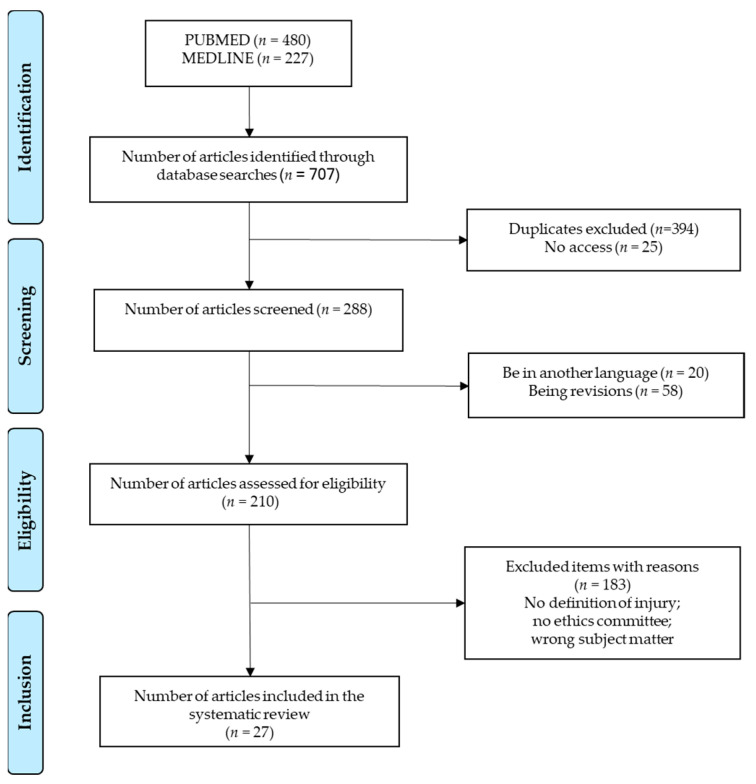
Flow diagram based on the PRISMA guidelines [11] of information flow through the different phases of a systematic review.

**Table 1 ijerph-19-10688-t001:** Criteria used to determine eligibility of studies.

Inclusion Criteria
(1) studies dealing with any handball injury, irrespective of gender (2) studies describing the pattern of injuries during training or competition in handball (3) any prospective study with reference to handball injuries at any age (4) studies showing the location of the different injuries in handball (5) studies on the prevention of any handball injury (6) studies that compile the injuries that have occurred during a particular competition (7) articles had to be originally published in English (8) they had to include the definition of an injury or a specific type of injury (9) they had to have passed an ethics committee or be a study that did not require ethics committee approval (10) studies with a minimum data sample size of 10 players (11) the date of publication had to be after 2012.
**Exclusion criteria**
(1) different topic from the one sought (2) sport other than handball (3) surveys (4) insufficient data (5) reviews (6) no definition of injury or of a specific type of injury is given (7) full text not accessible

**Table 2 ijerph-19-10688-t002:** Methodological quality of the selected studies [12].

	Raya-González et al. (2021) [10]	Mashimo et al. (2021) [14]	Asker et al. (2020) [15]	Goes et at. (2020) [16]	Luig et al. (2020) [17]	Palmer et al. (2020) [18]	Asai et al. (2020) [19]	Ruehlemann et al. (2019) [20]	Florit et al. (2019) [21]	Mónaco et al. (2019) [22]	Tabben et al. (2019) [23]	Aman et al. (2019) [24]	Aasheim et al. (2018) [1]	Oshima et al. (2018) [25]
Does the time period of the study appear?	Yes	Yes	Yes	Yes	Yes	Yes	Yes	Yes	Yes	Yes	Yes	Yes	Yes	Yes
Are there groups?	Yes	No	No	No	No	No	No	No	No	No	No	No	No	Yes
Does it talk about drop-outs or exclusions?	No	No	Yes	N/P	No	No	N/P	N/P	Yes	Yes	N/P	No	Yes	Yes
Does it talk about consent received from athletes?	Yes	Yes	Yes	Yes	Yes	Yes	Yes	Yes	Yes	Yes	Yes	Yes	Yes	Yes
Does it say what kind of study it describes?	Yes	Yes	Yes	Yes	Yes	Yes	Yes	No	Yes	Yes	Yes	Yes	Yes	Yes
Does it mention the parts of the body that are injured?	Yes	Yes	Yes	Yes	Yes	Yes	Yes	No	Yes	Yes	Yes	Yes	Yes	Yes
Mechanism of injury or situation that occurs (overuse, traumatic, contact or non-contact)?	Yes	Yes	Yes	No	Yes	Yes	No	No	No	Yes	No	No	Yes	Yes
Is the definition of injury or a specific type of injury given?	Yes	Yes	Yes	Yes	Yes	Yes	Yes	Yes	Yes	Yes	Yes	Yes	Yes	Yes
Is the average age of the subjects studied given?	Yes	Yes	Yes	Yes	No	Yes	Yes	Yes	Yes	Yes	Yes	Yes	Yes	Yes
Is the gender of the subjects studied given?	No	Yes	Yes	Y/N	Yes	Yes	Yes	N/P	Y/N	Yes	Yes	Yes	Yes	Yes
Is the number of injuries occurring or an injury incidence rate given?	Yes	Yes	N/P	Yes	Yes	No	No	No	Yes	Yes	Yes	Yes	Yes	Yes
Is time lost after injury discussed?	Yes	Yes	No	No	Yes	No	No	No	N/P	No	Yes	No	No	No
Severity of injury?	Yes	Yes	No	No	Yes	No	No	No	N/P	No	Yes	Yes	Yes	No
Differences in injuries between specific positions?	No	Yes	No	No	Yes	N/P	No	No	No	Yes	Yes	No	No	No
Points, No (%) (28 maximum)	22 (78.6)	24 (80.0)	19 (67,9)	16 (57.1)	22 (78.6)	19 (67.9)	17 (60.7)	10 (35.7)	19 (67.9)	22 (78.6)	23 (82.1)	18 (64.3)	22 (78.6)	22 (78.6)
	**Von Rosen et al. (2018)** [26]	**Asker et al. (2018)** [2]	**Andersson et al. (2018)** [27]	**Aman et al. (2018)** [28]	**Moller et al. (2017)** [29]	**Rafnsson et al. (2017)** [30]	**Giroto et al. (2017)** [31]	**Aman et al. (2016)** [32]	**Bere et al. (2015)** [33]	**Clarsen et al. (2015)** [34]	**Clarsen et al. (2014)** [35]	**Engebrestsen et al. (2013)** [5]		**Moller et al. (2012)** [3]
Does the time period of the study appear?	Yes	Yes	Yes	Yes	Yes	Yes	Yes	Yes	Yes	Yes	Yes	Yes		Yes
Are there groups?	No	No	No	No	Yes	No	No	No	No	No	No	No		Yes
Does it talk about drop-outs or exclusions?	Yes	Yes	Yes	No	Yes	Yes	Yes	No	No	Yes	Yes	N/P		Yes
Does it talk about consent received from athletes?	Yes	Yes	Yes	No	No	No	No	No	No	Yes	Yes	No		Yes
Does it say what kind of study it describes?	Yes	Yes	Yes	No	Yes	Yes	Yes	No	No	Yes	Yes	No		Yes
Does it mention the parts of the body that are injured?	Yes	Yes	Yes	Yes	Yes	Yes	Yes	No	Yes	Yes	Yes	No		Yes
Mechanism of injury or situation that occurs (overuse, traumatic, contact or non-contact)?	No	No	Yes	No	No	Yes	Yes	No	Yes	Yes	No	Yes		Yes
Is the definition of injury or a specific type of injury given?	Yes	Yes	Yes	Yes	Yes	Yes	Yes	Yes	Yes	Yes	Yes	Yes		Yes
Is the average age of the subjects studied given?	Yes	Yes	No	N/P	Yes	Yes	Yes	Yes	No	Yes	Yes	No		Yes
Is the gender of the subjects studied given?	Y/N	Yes	Yes	Yes	Y/N	Yes	Yes	Yes	Yes	Yes	Yes	Yes		Yes
Is the number of injuries occurring or an injury incidence rate given?	Yes	Yes	Yes	Yes	Yes	Yes	Yes	Yes	Yes	Yes	No	No		Yes
Is time lost after injury discussed?	N/P	No	No	No	No	Yes	Yes	No	Yes	No	No	Yes		Yes
Severity of injury?	N/P	No	No	Yes	No	Yes	Yes	Yes	Yes	No	No	Yes		Yes
Differences in injuries between specific positions?	No	Yes	No	No	No	Yes	No	No	Yes	No	No	No		No
Points, No (%) (28 maximum)	19 (67.9)	20 (71.4)	18 (64.3)	13 (46.4)	19 (67.9)	24 (85.7)	22 (78.6)	12 (42.9)	18 (64.3)	18 (64.3)	16 (57.1)	13 (46.4)		26 (92.9)

Legend: Two points for “Yes”, 1 point “N/P”, 0 points for “No”.

**Table 3 ijerph-19-10688-t003:** Characteristics of selected studies.

Study	Country	Aims	Status	Sample	Duration	Methodology	Results
Raya-González et al. (2021) [10]	Spain	To analyze the differences in professional handball players’ injury profile according to the team’s competitive-level (i.e., First division vs. Second division).	Elite	53 professional men’s handball players.	2015–2016 and 2016–2017 for the First division league and 2017–2018 and 2018–2019 for the Second division league.	A prospective cohort study (over four consecutive seasons).	No differences were found between the two teams. The second division team presented more injuries during training and a higher injury load than the first division team.
Mashimo et al. (2021) [14]	Japan	To reveal the injury profile based on player position in Japanese youth handball players.	Basic categories	2377 handball players.	Leagues 2018 and 2019.	Cross-sectional. Injury questionnaire.	There were significant differences in injury prevalence and patterns between player positions. The overall prevalence of injuries was 46.7%, with back players sustaining at least one injury higher than players in other positions.
Luig et al. (2020) [17]	Germany	To identify patterns and mechanisms of injury situations in men’s professional handball.	First or second league	1899 handball players.	Seasons from 2010 to 2013 and from 2014 to 2017.	Prospective study Video analysis	Contact injury is the most common, with wingers and pivots being the most affected. The most commonly injured areas are: knees (28.8%), ankles (20%), hands (11.7%) and thighs (9.5%). Injury risk is highest in the last ten minutes of each half of the game.
Asker et al. (2020) [15]	Sweden	To investigate whether players of both genders with shoulder muscle weakness, range of motion (ROM) rotation deficits, scapular dyskinesia in pre-season had a higher rate of new shoulder injuries compared to players without these characteristics.	Elite	452 handball players.	Season 2014–2015.	Prospective cohort study. The following were measured: shoulder strength, isometric external rotation, isometric internal rotation and eccentric rotation. Isometric abduction strength and scapular dyskinesia. OSTRC questionnaire on a weekly basis.	In the women handball players, an association was found between weakness between isometric external rotation and isometric internal rotation and the risk of injury. Association between scapular dyskinesia during glenohumeral abduction and risk of shoulder injury in the men handball players.
Goes et al. (2020) [16]	Brazil	To describe the prevalence and identification of factors associated with musculoskeletal injuries, including tendinopathy and joint and muscle injuries.		82 handball players Age: 25.2 years.	March 2018 to December 2018.	Cross-sectional observational study. Self-reported questionnaire.	Muscle injuries had the highest incidence followed by joint injuries and tendinopathies.
Palmer et al. (2021) [18]	International 131 countries	To describe the self-reported prevalence and nature of Olympic-career injury and general health and current residual symptoms in a self-selected sample of retired Olympians.	Retired Olympic	3357 retired Olympians.		Injury questionnaire.	Injury prevalence was highest in handball (82.2%).
Asai et al. (2020) [19]		To examine the incidence, type, and location of acute injuries in Japanese young handball players during national competition.	Basic categories	169 injuries were reported.	550 games from 2013 to 2018, held in March of each year.	Retrospectively assessed injuries.	The incidence of injuries in girls was lower than in boys. The greatest number of injuries were in the lower extremities, with the ankle being the most frequent. No statistical differences were found in the incidence of injuries by specific position
Florit et al. (2019) [21]	Spain	To describe the incidence and severity of tendinopathy in a multi-sport club with professional and youth teams.	Professionals and non-professionals	839 players (age: 8–38 years).	8 seasons (2008– 2016).	A retrospective epidemiological study. Classification with the Orchard Sports Injury Classification System (OSICS).	5.2% of professional players and 2.7% of young players had shoulder tendinopathy.
Ruehlemann et al. (2019) [20]	Germany	To objectively assess measures of knee joint stability with an established test battery in non-elite handball.	Non elite	165 handball players Age: 24.3 years.	-	Questionnaire. Battery of tests to measure balance, agility, speed and strength.	The data obtained can be used for an objective assessment of the functionality and stability of the knee.
Asker et al. (2018) [2]	Sweden	To assess the prevalence of shoulder problems among adolescent elite handball players to investigate possible differences in gender, school grade, playing position and level of play.	National and regional	471 handball players Age: 16.4 years.	Seasons 2014–2015 and 2015–2016.	Prospective cohort study. Baseline questionnaire and a modified Swedish version of the OSTRC.	Higher incidence of injuries in men and women front-line players, especially in the women.
Von Rosen, et al. (2018) [26]	Sweden	To describe injury patterns in terms of type: location, prevalence and incidence, recurrence and severity; time to first injury, in adolescent athletes and to compare differences in injury data by gender and sport type.	Elite	42 handball players Age: 17 years.	52 weeks.	Prospective cohort study. Weekly questionnaire (OSTRC and other questions). Injury history questionnaires (Questback online survey).	The most common and most severe injury was in the knee (17.2%). Most injuries occurred during competition and with no significant differences between genders.
Aasheim et al. (2018) [1]	Norway	To record overuse injuries among junior men players over the course of a handball season.	Elite	145 handball players Age: 17 years.	10 months.	Prospective cohort study. Baseline questionnaire. OSTRC questionnaire	Incidence of overuse injuries in young men particularly in the shoulder (higher average prevalence) and knee (higher relative load).
Aman et al. (2019) [24]	Sweden	To examine acute injuries in licensed floorball, football, handball and ice hockey players of all ages. To identify the most common and severe injuries at each body location and recommend average injury prevention.	Sportsmen and women from the Swedish Sports Confederation.	Total number of licensed players in the country.	From 2006 to 2015.	Observational study	The most common injury in both sexes is sprained/broken knee injury, followed by hand/finger fracture.
Oshima et al. (2018) [25]	Germany	To research the relationship between static balance and the incidence of non-contact anterior cruciate ligament injuries in high school athletes.	Handball at school	104 handball players Age: 15 years.	From April 2009 to 2011.	Prospective study. Postural sway is measured with a Gravicorder GS-31). Anthropometric data are collected.	Poor static balance may be a risk factor for non-contact ACL (anterior cruciate ligament) injury. Balance training effective in the prevention of ACL injuries in women.
Tabben et al. (2019) [23]	France	To study the association between players, characteristics, technical components of the game and the risk of injury during the men’s matches at the 2017 World Handball Championships.	Elite	387 handball players Age: 27.3 years.	11 to 29 January 2017.	Exploratory study An IOC injury and illness surveillance protocol using a methodology adapted for handball.	Most affected body parts in the men are: ankle, head/face, knee and thigh. Players more likely to be injured in front row. Older players at higher risk of injury than younger players.
Mónaco et al. (2019) [22]	Spain	To estimate the influence of position, category and maturity stages on the incidence and pattern of injuries in handball players.	Elite	164 handball players Age: 15.5 years.	Seasons 2011–2012 and 2012–2013.	Cohort study. Medical examination. Classification of injuries using OSICS-10.	In men, the most frequently injured areas are the ankle, knee and thigh. Most common injuries are ligament/joint sprains and muscle strains. The second line has higher risk of knee injury. Incidence is higher in matches than in training.
Møller et al. (2017) [29]	Denmark	To research whether an increase in handball load is associated with higher rates of shoulder injury compared to a smaller increase or decrease, and whether the association is influenced by scapular, isometric shoulder control.	Elite	679 handball players Age:16 years.	31 weeks 13 October 2013 to 11 May 2014.	Cohort study. Assessment of isometric shoulder rotation, abduction strength, ROM and scapular control at baseline and mid-season. Injury monitoring by SMS, telephone and the medical examination surveillance system (SPEx).	The number of injuries increased with a 60% increase in training load. Scapular dyskinesia and reduced rotational strength increased the risk of shoulder injury with a 20% increase in training load.
Rafnsson et al. (2017) [30]	Iceland	To examine the incidence, type, location and severity of injuries in Icelandic elite handball players and compare across factors such as physical characteristics and playing position.	Elite	109 handball players Age: 23.4 years.	Season 2007–2008.	Prospective cohort study	A high number of acute injuries, mainly in the lower extremities. The most frequent areas of injury were the lumbar region or pelvis, the knee, shoulders and sprains. Sixty-four percent of the injuries were non-contact. Higher incidence of injuries in goalkeepers, followed by fullbacks and forwards.
Andersson et al. (2018) [27]	Norway	To assess whether previously identified risk factors are associated with the overuse of shoulder injuries in men and women players.	Elite	329 handball players.	From October 2014 to March 2015.	Prospective cohort study. Baseline questionnaire. Measurement of external rotation, internal rotation and ROM, isometric strength of internal and external rotation on the dominant side. Measurement of scapular control. Recording of shoulder problems through the OSTRC.	A possible risk factor for overuse shoulder injury was identified as gender. The women had a higher prevalence of shoulder problems.
Åman et al. (2018) [28]	Sweden	To identify which injuries to focus prevention efforts on in order to have a significant impact on reducing acute injuries nationally.	National level	130,573 handball players.	2006–2013.	Data collected from the Swedish insurance company Folksam.	Increased internal rotation ROM was significantly associated with overuse shoulder injury.
Aman et al. (2016) [32]	Sweden	To identify high-risk sports with respect to reported acute injury incidence and injury severity in 35 sports.	National level	All ages.	2008 to 2011.	Insurance company details.	Upper and lower extremities were at high risk of injury and trunk/back at minimal risk in both genders. Lower extremities had a higher proportion (higher risk in women) compared to upper extremities.
Bere et al. (2015) [33]	Qatar	To describe the pattern of injuries and illnesses at the 2015 Men’s Handball World Championship.	Elite	384 handball players.	From 15 January to 1 February (18 days).	Internal Olympic Committee (IOC) surveillance system.	Lower extremities accounted for 58.3% of injuries (mainly ankle, thigh and knees). Of the injuries, 61.4% were contact injuries and more frequent during the second part of the first half. Highest incidence of injury to players on the 6-m line, followed by wingers, front row and goalkeepers.
Giroto et al. (2017) [31]	Brazil	To investigate the incidence and risk factors for handball injuries in elite Brazilian handball.	Elite	339 men and women handball players Age: 23.4 years.	From May to November 2011	Prospective cohort study. Baseline questionnaire.	The most common injuries of traumatic origin were ankle and knee. The most common overuse injuries were shoulder and knee. Tendinopathy was the most recorded. The majority of injuries, were of moderate severity (2–7 days).
Clarsen et al. (2015) [34]	Norway	To describe the extent of overuse problems in five different sports.	Elite	55 handball players.	13 weeks	Prospective cohort study. Weekly questionnaire.	High incidence of overuse injury in the knee and shoulder.
Clarsen et al. (2014) [35]	Norway	To determine whether rotator cuff strength, glenohumeral joint range of motion and scapular control are associated with shoulder injuries.	Elite	206 handball players Age: 24 years.	S30 weeks.	Prospective cohort study. Baseline questionnaire. Glenohumeral internal and external rotation was measured. ROM was measured. Isometric strength of internal and external rotation and abduction. Assessment of scapular control. OSTRC questionnaire.	Internal risk factors associated with shoulder injury.
Engebrestsen et al. (2013) [5]	England	To analyze injuries and illnesses that occurred at the London 2012 Olympic Games.	Elite	349 handball players.	24 July to 12 August 2012.	IOC injury and illness surveillance system. Daily reporting of injuries in a standardized way.	A greater number of injuries have occurred during competition than during training. Women players had a total of 26.3% injuries with 5.8% with injuries with more than 7 days lost. Men players had 17.4% of injuries with 3.4% with injuries with more than 7 days lost.
Moller et al. (2012) [3]	Denmark	To assess the incidence of injuries in elite handball and whether gender and previous injuries are a risk factor for new injuries.	Elite	517 handball players Ages: Under 16 and under 18.	From September 2010 to April 2011.	Cohort study. Baseline questionnaire. Weekly injury.	Most injuries were traumatic in origin and the remainder were from overuse. Sixty-five per cent of the injuries were new and 35% were recurrent. For the U16 group a previous injury was a significant risk factor for injury. Moderate injuries had the highest incidence.

## Data Availability

Not applicable.

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
