# Peer review of "The Most Common Handball Injuries: A Systematic Review"

_ijerph, 2022, doi:10.3390/ijerph191710688_

Round 1

Reviewer 1 Report

Review 

The Most Common Handball Injuries: A Systematic Review

ijerph 1862420

The paper analyses 27 materials related to the injuries suffered during the handball game considered as the Olympic game with the highest rate of injuries. The analysis has a multi-sided approach referring to: the players’ gender, the position in the game, the most often injured body parts, the moment of injury. The analysed material can be useful for trainers, in order to plan the training tasks in order to prevent injuries.

The paper is well structured/organised containing: introduction, materials and methods, results, discussions, conclusions. Each of the composing parts is detailed, the authors succeeding in synthesizing the specialised literature they had in view. In the paper, the objective of the study is outlined, namely that of identifying, placing and comparing the most frequent injuries and injury mechanisms in the practice of handball.

As for the research methodology, it is explained clearly and in detail, besides the strategy of searching for relevant materials, the criteria considered by the authors for their selection are indicated, as are the data collection and the methodological design. The data collected from the 27 selected articles are interpreted adequately and consistently throughout the paper. 

The bibliographical references are mostly latest publications ( issued in the last five years) and relevant for the studied topic. The tables are easily to be interpreted and understood.

The paper is well written and easy to be understood, which offers me the certainty that the results presented will be useful both for trainers and players, especially professional ones, in their demarche to avoid as much as possible, the injuries eitherduring trainings or during competitions.

Despite all these I could neither identify the place and role of epidemiology” (key term) in the article, nor ”prevention”(key term) is presented in discussions or in conclusions.

The information taken from all 27 articles are analysed, correlated which does not justify the detailed presentation of each article.

Author Response

Dear reviewer,
Thank you for reviewing our manuscript. We have carefully considered all the considerations in the document you have provided. Please find attached in the word document our detailed answers to your questions.

Reviewer 2 Report

The authors claim to use the PRISMA statement in the research, which isn’t well defined by the cited reference in the current paper. Also, the authors should provide more accurate references in the context of the paper content.

How were the selection criterions (inclusion and exclusion) chosen? There isn’t no support for the items related in the Table 1. The statement “The search and selection process of the articles was carried out by one researcher, who determined whether the studies met the defined inclusion and exclusion criteria” isn’t well defined. A scientific paper must have solid connections with other scientific references in the literature.  

Only 27 studies found in the literature for a systematic review represents a low number. There are research papers that aren’t indexed in the mentioned databases (PUBMED and MEDLINE), so must be consider all the known scientific databases.

Table 2 isn’t appropriate in this format for the paper. Why the selection questions were taken 1 to 1 from other scientific reference,” Analysis of Injury Incidences in Male Professional Adult and Elite Youth Soccer Players: A Systematic Review”? The authors must define their own selection criterion for the systematic review.

Table 3 must be removed or must be re-organized for a better and simple view of the characteristics. Some paragraphs from the table are copied from the source document and must be rephrased, otherwise will be taken as plagiarism.

Some graphs can be useful for the reader to view the results obtained in the review process. Maybe some statistical analysis can be provided with this study.

Overall, in the scientific literature are many reviews regarding the injuries met in the handball game, so the scientific content for this paper is poor. What is this study better than others? Is more complex? Please provide more arguments why this study is appropriate to be published in the journal IJERPH.

Author Response

(The authors gave the same response as above.)

Reviewer 3 Report

First of all, congratulations on this research article in which you are approaching the traumatology area of the handball game.

Thus it is well structured, I would consider putting in the discussion area information about prevention since you approached it in the abstract and also selected the term as a keyword.

Author Response

(The authors gave the same response as above.)
